# The Relationship Between Well-Being and MountainTherapy in Practitioners of Mental Health Departments

**DOI:** 10.3390/ijerph22081181

**Published:** 2025-07-25

**Authors:** Fiorella Lanfranchi, Elisa Zambetti, Alessandra Bigoni, Francesca Brivio, Chiara Di Natale, Valeria Martini, Andrea Greco

**Affiliations:** 1ASST Bergamo Est, 24068 Seriate, Italy; fiorella.lanfranchi@gmail.com; 2Italian Society of Mountain Therapy, 35128 Padova, Italy; 3Human and Social Sciences Department, University of Bergamo, 24129 Bergamo, Italy; elisa.zambetti@unibg.it (E.Z.); francesca.brivio@guest.unibg.it (F.B.); andrea.greco@unibg.it (A.G.); 4Ecolab, 20871 Vimercate, Italy; 5Psychiatric Day Center, Mental Health Service Sant’Egidio alla Vibrata (TE), Mental Health Department, ASL4 Teramo, 64016 Sant’Egidio alla Vibrata, Italy; dinatalechiara91@gmail.com (C.D.N.); vmartini@libero.it (V.M.)

**Keywords:** work-related stress, burnout, mountain group activities, psychological well-being

## Abstract

*Background*. Healthcare workers’ health can be influenced by physical, psychological, social, emotional, and work-related stress. MountainTherapy Activities (MTAs) are an integrated therapeutic approach that uses nature to enhance their well-being through group activities like hiking. This cross-sectional study examines well-being levels among Italian Departments of Mental Health workers who do or do not participate in MTAs. It hypothesizes that MTAs may reduce burnout, boost psychological resilience, and increase job satisfaction. *Methods*. The study involved 167 healthcare workers from 11 Italian Local Health Authorities, divided into MTA (who take part in MTA; *n* = 83) and non-MTA (who have never participated in MTA; *n* = 84) groups. They completed five validated questionnaires on psychological distress, burnout, resilience, job engagement, and psychological safety. Data were compared between groups, considering MTA frequency and well-being differences during MTAs versus workplace activities. *Results*. MTA participants scored higher in psychological well-being (*t*(117.282) = −1.721, *p* = 0.044) and general dysphoria (*t*(116.955) = −1.721, *p* = 0.042). Additionally, during MTAs, they showed greater job engagement (vigor: *t*(66) = −8.322, *p* < 0.001; devotion: *t*(66) = −4.500, *p* < 0.001; emotional involvement: *t*(66) = −8.322, *p* = 0.002) and psychological safety (general: *t*(66) = −5.819, *p* < 0.001; self-expression: *t*(66) = −5.609, *p* < 0.001) compared to other activities. *Conclusions*. MTAs can be considered a valid intervention for the promotion of the mental health of healthcare workers.

## 1. Introduction

### 1.1. Work-Related Stress and Burnout in Healthcare Services

According to widely accepted interpretative models, work-related stress results from a distorted interaction between the demands (physical, cognitive, emotional, relational) expected by a task and/or role and the individual’s ability to manage these demands, reflected in their psychophysiological, behavioral, and operational responses [1,2,3,4,5]. Work-related stress can significantly impact an employee’s engagement, performance, and productivity, ultimately leading to negative outcomes for the organization [6,7,8]. Several studies [9,10,11,12] suggest that extended durations of stress can substantially jeopardize an individual’s health. Consequently, work-related stress is a critical factor to consider because chronic stress can lead to burnout. Burnout [13] involves rapid emotional and physical exhaustion and decreased work engagement. Due to chronic stress, burnout primarily affects individuals in caregiving roles [14,15]. In 2022, the World Health Organization (WHO) officially classified it as an occupational phenomenon in the International Classification of Diseases (ICD-11) [16].

Healthcare professionals are particularly vulnerable to this risk due to their work environment, which includes nursing homes, mental health departments, and other healthcare settings. In these locations, they regularly interact with individuals who require health, social, and psychological support because of various forms of fragility or illness, and this can result in significant job-related stress. This distress can adversely affect their well-being and increase clinical risks associated with unrecognized and unmanaged stress [15,17], for example, risk of developing mental health problems (like anxiety disorders, depression disorders, and PTSD), distress, and burnout related to precisely these stressors [18,19]. The literature [20,21,22,23,24] suggests that the highest levels of stress and the development of stress-related disorders, burnout, and physical or psychiatric conditions are experienced by nurses [20,21], women [22], older people [22], and people with a longer length of service [23,24].

### 1.2. Theoretical Models and Psychological Well-Being in the Work Context

Karasek’s model [3], later expanded by Johnson [25], identifies three factors in work-related stress. The first, job demand, refers to the psychological, physical, social, and organizational aspects of work that incur psycho-physical costs. Secondly, job control represents personal freedom in decision-making, crucial for achieving goals and reducing costs while promoting personal growth. Thirdly, job support encompasses the assistance received in the workplace, which helps alleviate stress and enhances the perception of capability and control. High stress arises when there is a significant gap between job demands and worker resources, combined with low job control and low job support [2,3,6,25].

The focus of research has transitioned from assessing the negative aspects of the psychosocial work environment, commonly referred to as negative stress, to examining the overall level of well-being within the workplace. This is described as “the set of cultural cores, processes, and organizational practices that invigorate the dynamics of coexistence in work contexts, promoting, maintaining, and enhancing the quality of life and the level of physical, psychological, and social well-being of working communities” [26]. Both individual employees and health organizations can therefore develop processes of resilience and empowerment that employees can benefit from to balance perceived stress levels. Well-being benefits could derive from natural areas [27,28,29,30,31] or exposure to green spaces [29,32]. Exposure to natural settings may therefore contribute to reducing stress and fostering a positive psychological state, generating positive physiological and psychological responses [33]. In addition, the psychological benefits of natural experiences may be strengthened thanks to mindfulness and mind wandering [34]; decentering and deliberate mind wandering were associated with positive outcomes, including stronger nature connection and positive affect [34,35]. These studies support the potential value of natural environments as settings for the improvement in healthcare delivery and therapeutic interventions. Confirming these results, a recent study by Joscko and colleagues (2023) [36] analyzes relationships between nature-based therapy, mental health, and individuals’ connectedness to nature in a young psychosomatic sample. The findings suggest improvements in mental well-being and show a stronger sense of connectedness to nature as a result of the therapy, as well as a reduction in depression scores. Moreover, the whole sample considers nature-based therapy as effective.

### 1.3. Environmental and MountainTherapy Activities

The main aim is to verify whether the activities of MountainTherapy represent an experiential opportunity that can promote the physical, psychological, and social well-being not only of patients but also of professionals, thereby serving as a protective factor against burnout and promoting psychological well-being even in the workplace. Ryff and Keyes (1995) [37] described psychological well-being as a multidimensional concept that includes several key elements like self-acceptance, positive relationships, environmental control, autonomy, or purpose in life. These factors are crucial for maintaining a healthy psychological state. MountainTherapy was developed as a unique methodological approach aimed at individuals with psychological illnesses or disabilities, set in the cultural, natural, and artificial environments of the mountains [38]. Currently, these activities are primarily designed and carried out within the National Health Service, with crucial collaboration from the Italian Alpine Club (CAI) and other accredited organizations and associations in the field, and in Italy, there are approximately 1600 health and social workers involved in MountainTherapy activities (MTAs). MountainTherapy experiences have also been known in other European countries, such as France, Belgium, and Spain, since the early 1980s, in various fields such as psychiatry, addiction, and disability services [39]. A key factor that differentiates MountainTherapy from other therapeutic interventions conducted inside or outside clinical settings is the use of the mountain environment as a developmental, change, and rehabilitation tool. Exposure to the mountain and natural environment brings added value by performing a regenerative function related to immersion in greenery. It promotes greater sensory awareness, potential for physical and emotional growth, and the achievement of a flow state, as indicated by research conducted by the University of Milan [40]. That study shows how patients with schizophrenia who participate in MTAs can experience a positive state of flow of consciousness, maintaining it for longer periods compared to other rehabilitation activities. Another defining factor of MountainTherapy is physical exercise, which is a powerful supplement to existing standard treatments. Physical training, when used in conjunction with other therapeutic, pharmacological, and psychosocial approaches, can bring much more significant improvements in mental states and quality of life [41] as well as being a neuroprotective factor [42]. Some studies indicate that the greatest benefits for cognition and the course of schizophrenia at onset are given by the combination of aerobic exercise and nonpharmacological interventions [43]. With MountainTherapy, repeated global experiences are constructed using body meditation to stimulate a reintegration of the self into the self and a resumption of relational skills. The outdoor journey is shared in groups. Precisely, the group is another of the characterizing dimensions of MountainTherapy projects. The group is made up of patients, caregivers, volunteers, and technical mountain experts. This is an action-centered therapy and concrete experience, in which the patient becomes an active subject, involved in multidimensional interactions (physical, affective, social, and cognitive). It represents an unfamiliar experience that is “at odds” with the patients’ usual reality and can foster openness, and the acquisition of new perspectives of the world and the self. It requires enacting adaptive behaviors and focusing on one’s abilities rather than one’s dysfunctions [40]. “Doing together” involves moments of exchange that require clinical attention. The group is a mediator that provides space to develop sustainable interpersonal relationships that help individuals experience a sense of belonging. As indicated in the research of Scala and colleagues (2006) [44], the MountainTherapy experience leads to greater satisfaction with living conditions, provides experiences critically absent in the existence of many psychotic patients, helps to overcome isolation and feel part of the group, to be within an evolutionary process of improving skills and competencies, and to develop interest and curiosity in communicable and shareable domains of activity. MountainTherapy sessions include a part of concrete action, mainly hiking activities, to which climbing, cross-country skiing, and “snowshoeing” are added. In addition to the motor activity, group sessions are held during the outing for reflection and discussion on specific issues. Particular work is undertaken on communication, social interaction, and leisure management skills, to help broaden the patients’ ability to read experience and strengthen their social functioning. Social skills training [45] is initiated within psychiatric facilities and continued in real-life settings, such as mountains, promoting the generalization of stimuli and responses. After the outings, the lived experience is reframed both in group meetings and individually in sessions with the therapist. The risks in performing MountainTherapy are related to frequenting the mountain environment. To date, no major accidents or injuries have been reported to either operators or patients during the outings carried out over the past few decades and improve cognitive functioning [46]. The activities are planned and implemented mainly within the framework of the National Health Service, with the fundamental collaboration of the Italian Alpine Club and other accredited organizations or associations in the field. Each group includes Italian Alpine Club instructors and volunteers or professional mountain guides, who have the specific task and responsibility of taking care of the participants’ safety. In any case, if the operator does not already have personal knowledge of frequenting the mountains, they must be trained tailored to the specific conditions they may encounter. This training equips them with the knowledge and skills necessary to navigate safely and effectively in a mountain environment. Technical training should be understood in the individual’s ability to interpret the mountain, with an awareness of useful strategies, both in managing the geographical and meteorological environment and in dealing with the various activities offered. One of the fundamental and unique aspects of MountainTherapy is the very fact that it steps outside the walls of institutions. The mountain, as a variable natural environment, leads to a potential deconstruction of the traditional setting, with a partial breaking down of hierarchies between patients and practitioners that are almost impossible to see in other groups. During a hike, operators, often seen by patients as omnipotent figures, show their physical limitations and fatigue, which are sometimes like those of the users themselves. Being open to the outside world and joining a group in which Alpine Club volunteers participate makes patients feel that they are an integral part of the group of hikers who do not have mental illness since they share with them the same interest and passion for the mountains. This allows patients to get out of a context related to dependence on mental health services and hang out with non-sick people at the same time. The operator plays the role of a “relationship facilitator”, a mediator between users and mountain experts, a trait union between the experiences lived in institutional and open places [37,39].

The literature shows that mountain environments act as revitalizing natural settings that support psychological well-being, with both immediate and lasting benefits [39]. Specifically, mountain sport activities such as mountaineering/alpine climbing, climbing, mountain hiking, skiing, and mountain biking have the potential to enhance an affective state. As explained in attention restoration theory (which is a framework that analyzes which environmental characteristics enhance directed attention), mountain environments may help restore cognitive resources [39]. A study comparing the restorative impact of interacting with natural versus urban environments on cognitive functioning by Berman and colleagues (2008) [47] aligns with these results. In fact, natural settings, especially the ones rich in softly fascinating stimuli, engage attention in a bottom-up manner, thereby allowing top-down attentional mechanisms to recover. In particular, findings from two experiments included in the study demonstrate that both walking in nature and viewing nature images significantly improve directed attention. Based on that, it is evident that the literature reveals a gap in exploring the influence of nature on the well-being of health professionals.

### 1.4. Aims of the Study

The study seeks to investigate the well-being levels of health professionals employed in Italian Departments of Mental Health who regularly engage in MTAs and those who do not. Furthermore, it hypothesizes three different objectives: (objective a) professionals involved in MountainTherapy activities exhibit higher psychological well-being, resilience, psychological safety, engagement, and well-being during work activities and lower levels of burnout risk compared to those who do not practice these activities; (objective b) the frequency of these activities can positively influence the psychological well-being levels of participants; and (objective c) the MT Group will achieve higher well-being scores when MT activities are compared to other workplace activities (e.g., undergoing therapy and visits patients in the facility).

## 2. Materials and Methods

### 2.1. Design and Sampling

The present study has been designed as cross-sectional, allowing the investigation in a specific moment in time. Two groups of healthcare professionals were involved in the study, affiliated with the Mental Health Departments of several Italian ASLs. They were divided based on their participation (MountainTherapy Group—MT) or non-participation in MTAs (No-MountainTherapy Group—nMT). Participants are all those who join the MTA programs offered by their department, with varying frequency: once a week, once every two weeks, once a month, or participation in visits. Participants were recruited through connections with Italian Mental Health Departments involved in ongoing MountainTherapy projects. This multicenter research was initiated by the Italian MountainTherapy Society in collaboration with the University of Bergamo. The Ethical Committee of this university approved the study.

Participants were recruited through convenience sampling. Participating departments were requested to identify all healthcare professionals who engaged in MountainTherapy activities (MT Group) during 2023, as well as another group of professionals who did not perform MountainTherapy but belonged to the same services and shared similar characteristics as the participants (nMT Group). Data collection took place between April and May 2023. In relation to the sample size calculation, in order to achieve a significance level of *p* = 0.05, a power of 80% (1−β = 0.80), and to detect medium effects (d = 0.50), the required sample size was at least 128 participants. A total of 167 participants participated in the research (their sociodemographic characteristics are shown in Table 1), evenly distributed between the MT Group (*n =* 83; 49.7%) and the nMT Group (*n =* 84; 50.3%).

### 2.2. Procedure

Participants from both groups filled out five self-report questionnaires via the Qualtrics platform. Participants in the MT Group also completed two additional questionnaires in a modified form regarding the specific MT activities. This decision was made in response to objective c, to test whether the higher scores within the MT Group are obtained when MTAs are considered compared to other work activities.

At the end, participants were asked to provide some sociodemographic information, such as age, gender, geographic location of the afferent facility, affiliated service, role in the team, how long they have been working in the service, and only for participants involved in MTAs, how long they have been engaged in these activities (less than 1 year, 1 to 3 years, 3 to 5 years, 5 to 10 years, or more than 10 years) and the frequency of participation (once a week, one time every two weeks, once a month, or participation in visits).

### 2.3. Questionnaires

#### 2.3.1. General Health Questionnaire [GHQ-12], Italian Version

The General Health Questionnaire [GHQ-12] [48], the Italian version of Piccinelli and Politi [49], has been designed to identify potential warning signs regarding psychological distress. The questionnaire consists of 12 items related to how the person has felt over the last two weeks, rated on a 4-point Likert scale (from 0 = much less than usual to 3 = more than usual). It is possible to calculate both a mean score and a total score; the higher the score, the higher the individual’s psychological distress. Additionally, a “traffic light” system for overall psychological distress and well-being can be created [50]: green (score 0–14) indicates a good psychological state (and so lower psychological distress); yellow (score 15–19) indicates a borderline range, at risk of experiencing psychological distress; and red (score 19–36) indicates potential psychological distress (like anxiety, depression, or emotional disorders), which should be attended to from a clinical point of view. In addition, this instrument consists of two subscales, for which respective scores can be calculated: the scale related to the dimension of general dysphoria and the scale related to social functioning [49]. The first subscale (general dysphoria) relates to psychological distress related to emotional symptoms such as anxiety, depression, and general dissatisfaction (sample items: “*feel tense*”; “*feeling unhappy or depressed*”); the second subscale (social functioning) reports practical difficulties in managing daily life and maintaining efficiency and relationships (sample items: “*be able to concentrate*”; “*enthusiastically carry out daily activities*”).

Internal consistency of the instrument is excellent for the total score (Cronbach’s α = 0.88), good for the dimension of general dysphoria (Cronbach’s α = 0.87), and discrete for the dimension related to social functioning (Cronbach’s α = 0.77).

#### 2.3.2. Maslach Burnout Inventory [MBI], Italian Version

The Maslach Burnout Inventory [MBI] [51], the Italian version of di Sirigatti and Stefanile [52], is aimed at investigating burnout syndrome. It consists of 22 items for which the respondent must evaluate—on a 7-point Likert scale (from 0 = never to 6 = daily)—the frequency and intensity with which they experience symptoms, effects, and emotional states related to their work, attributable to three independent dimensions of burnout syndrome: emotional exhaustion, which is the feeling of emotional detachment and disinterest towards one’s work and others (sample items: “*I feel emotionally exhausted by my work*”; “*I think I work too hard*”); depersonalization, which involves demonstrating detached and impersonal responses and behaviors, sometimes even cynical (sample items: “*since I started working here I have become more insensitive to people*”; “*I am afraid that this work will harden me emotionally.*”); and personal accomplishment, which refers to the perception of one’s competence and desire for success in working with others, which, in cases of burnout, can manifest as a decrease in self-esteem due to a perception of oneself as not adequately performing in the work context (sample items: “*I believe that I am positively influencing other people’s lives through my work*”; “*I have accomplished many valuable things in my work*”).

Internal consistency of the instrument is excellent for the dimension of emotional exhaustion (Cronbach’s α = 0.89), sufficient for the dimension of depersonalization (Cronbach’s α = 0.66), and discrete for the dimension of personal accomplishment (Cronbach’s α = 0.74).

#### 2.3.3. Utrecht Work Engagement Scale [UWES-9], Italian Version

The Utrecht Work Engagement Scale [UWES-9] [53], the Italian version of Balducci et al. [54], consists of 17 items for which the participant must assess—on a 7-point Likert scale (from 0 = never to 6 = daily)—the frequency with which they have experienced certain sensations and/or feelings related to their work, concerning three dimensions: vigor, which refers to having a high level of energy and endurance, especially mental (sample items: “*in my work I feel strong and vigorous*”; “*in my work I am always persevering even when things are not going well*”); devotion, which means feeling like an important part of the work one is engaged in, finding inherent meaning in one’s work (sample items: “*I find the work I do rich in meaning and purpose*”; “*my work inspires me*”); and involvement, which refers to concentration, the time devoted, and being immersed in one’s work (sample items: “*when I work, I forget about everything else*”; “*I am immersed in my work*”). Participants in the MT Group were also given this scale in a modified version that solely pertained to activities related to MountainTherapy, replacing the term “work” with “activities” (sample items: “*during activities I feel strong and vigorous*”; “*during activities I am always persevering even when things are not going well*”; “*I find the activities I do rich in meaning and purpose*”; “*activities inspire me*”; “*when I do activities, I forget about everything else*”; “*I am immersed in activities*”).

We calculated the internal consistency in the two groups (MT and nMT), for each of the three dimensions. Concerning the MT Group, the internal consistency of the instrument is excellent for the dimension of vigor (Cronbach’s α = 0.93), devotion (Cronbach’s α = 0.96), and involvement (Cronbach’s α = 0.89). Concerning the nMT Group, the internal consistency of the instrument is good for the dimension of vigor (Cronbach’s α = 0.83), excellent for the dimension of devotion (Cronbach’s α = 0.93), and good for the dimension of involvement (Cronbach’s α = 0.79).

#### 2.3.4. Resilience Scale [RS-14], Italian Version

The Resilience Scale [RS-14] [55], the Italian version of Callegari et al. [56], is designed to investigate individuals’ resilience and the resources they have available to effectively cope with stressful or traumatic events, face difficulties, and adapt to change. Participants are asked to indicate their level of agreement—scoring on a 7-point Likert scale (from 1 = strongly disagree to 7 = strongly agree)—with each of the 14 items that make up the scale (sample items: “*I feel proud of the things I have accomplished in my life*”; “*in an emergency situation there is someone I can count on*”), resulting in a total score that ranges from a minimum of 14 to a maximum of 98. A score below 56 indicates a very low level of resilience; a score between 57 and 64 indicates a low level of resilience; a score between 65 and 73 indicates that the level of resilience is on the borderline of being considered low; a score between 74 and 81 indicates a moderate level of resilience; a score between 82 and 90 indicates a moderately high level of resilience; and a score above 91 indicates a high level of resilience.

Internal consistency of the instrument is excellent (Cronbach’s α = 0.94).

#### 2.3.5. Edmondson’s Team Psychological Safety Scale [57]

The Edmondson’s team psychological safety scale [57], the Italian version will be published by Todaro et al. [58], is a scale that measures the psychological safety perceived by individuals within their work context—through responses to 7 items on a 7-point Likert scale (from 1 = strongly disagree to 7 = strongly agree)—indicating how safe individuals feel in interpersonal relationships that develop during work activities and within teams (sample items: “*I feel free to be completely myself at work*”; “*the emotions I express at work are my true emotions*”). Participants in the MT Group were also given this scale in a modified version so that it specifically referred to the activities related to MountainTherapy, replacing the term “work” with “activities” (sample items: “*I feel free to be completely myself during activities*”; “*the emotions I express during activities are my true emotions*”).

We calculated the internal consistency in the two groups (MT and nMT), for both dimensions. Concerning the MT Group, the internal consistency of the instrument is sufficient for the dimension of general psychological safety (Cronbach’s α = 0.60), and good for the dimension of self-expression psychological safety (Cronbach’s α = 0.78). Concerning the nMT Group, the internal consistency of the instrument is discrete for the dimension of general psychological safety (Cronbach’s α = 0.80), and good for the dimension of self-expression psychological safety (Cronbach’s α = 0.87).

### 2.4. Statistical Data Analysis

The analyses were conducted using SPSS software Version 29.0.1.0. Descriptive statistics were performed to analyze the sample distribution and sociodemographic variables for the total sample and within the two groups (MT and nMT). The dimensions of the scale used for the study were preliminarily submitted to analyses to check the normal distribution by calculating the mean, standard deviation, and indices of skewness and kurtosis; West et al. (1995) [59] recommend concern if skewness > |2| and kurtosis > |7|. Subsequently, independent *t*-tests were conducted to assess the effect of participating or not participating in MTAs (objective a) on the different psychological scales outlined above. The same test has been used to compare the values of the various psychological dimensions in relation to the frequency of these activities (objective b). Finally, a dependent *t*-test has been used to investigate in the MT Group only (objective c) the mean scores on key psychological scales regarding engagement and personal well-being as well as psychological safety, comparing the values in the work context and in MTAs.

## 3. Results

A total of 167 healthcare operators participated in the study from the Mental Health Departments of the Local Health Authorities. The majority of participants in the study are female, representing 74.2% in the MT Group and 70.8% in the nMT Group. The average age of participants in the MT Group is 44.67 years (*SD* = 11.32), while for the nMT Group, it is 47.1 years (*SD* = 9.86). The professional backgrounds of the involved participants include outpatient, residential, and semi-residential services, primarily located in North-East Italy. In terms of years of experience, most participants in the MT Group have been working in their role for less than 5 years or over 20 years; conversely, in the nMT Group, the majority have been in their positions for at least 10 years. Both groups predominantly consist of professional educators and psychiatric rehabilitation therapists collectively. Additional details about the participants’ sociodemographic characteristics can be found in Table 1.

First of all, considering the West et al. (1995) [59] recommendations, all the dimensions showed an acceptable distribution (Skewness_MIN_ = −0.82—Skewness_MAX_ = 1.22; Kurtosis_MIN_ = −0.96-Kurtosis_MAX_ = 2.56).

The bivariate analyses showed that the MT Group and the nMT Group did not exhibit a statistically significant association for any of the following variables: gender, age, affiliation service, geographical origin, length of service, and role.

Concerning the variables of interest, we observed several statistically significant associations and noteworthy trends towards significance, both in the comparison between the two groups and within the MT Group itself. Comparing the two groups (objective a), it was found that in psychological distress—measured using the GHQ-12 scale—those who do not participate in MTAs have, on average, higher scores both at the total level and in the sub-dimension of general dysphoria. It is important to note that a higher average score in these dimensions corresponds to a higher level of psychological distress. As shown in Figure 1, significant differences emerge in the distribution of the two groups across the three traffic lights (*χ*2(2, *n* = 136) = 6.398, *p* = 0.041): most of the participants positioned in the green light category, indicating that they do not exhibit psychological problems related to distress, are those who took part in MTAs (*n* = 30; 61.2%). Conversely, among the operators who fall into the borderline category of discontent, the majority are those who do not engage in MountainTherapy (*n* = 37; 62.7%).

Table 2 and Table 3 show the comparison between the MT and nMT groups and within the MT Group, regarding the psychological dimensions evaluated.

Considering objective a and regarding the burnout dimensions—assessed using the MBI—a difference approaching significance is noted only in the average scores of the dimension related to work accomplishment, which is found to tend to be significantly higher among those participating in MTAs compared to those who do not participate. Analyzing engagement and personal well-being at work using the UWES-9 scale, a trend towards significance emerged from the comparison between the two groups: participants in MountainTherapy tend to have higher average scores than non-participants in the dimension related to vigor. This suggests that participating in MTAs generally leads to better engagement and personal well-being at work in terms of vigor.

Regarding resilience in the workplace context—RS-14—questionnaire, there were no statistically significant differences between the average scores of participants in the MT Group (M = 76.72, SD = 13.86) and non-MT Group (M = 77.12, SD = 13.30); participating or not in MTAs does not seem to influence the average resilience of individuals in their work context, and subjects from both groups fall within a moderate level of resilience, with scores ranging from 74 to 81. Furthermore, when analyzing the different resilience ranges in more detail, no statistically significant differences emerged in the distribution of the two groups: the majority of both MountainTherapy participants and non-participants—62.4% and 67.7%, respectively—are positioned in the resilience range from moderately high to high, indicating that all participants exhibit good resilience characteristics.

Similarly, there are no statistically significant differences between participants and non-participants in MTAs regarding the average scores related to psychological safety [60] in the work context, both in its general dimension (MT Group: M = 5.01, SD = 1.19; nMT Group: M = 5.01, SD = 1.06) and in the dimension related to self-expression (MT Group: M = 5.35, SD = 1.09; nMT Group: M = 5.20, SD = 1.00).

Related to objective b of this study, we have analyzed more deeply these variables within the MT Group only, to compare the average scores only during MTAs concerning the time spent on these activities. Statistically significant differences were found in the three burnout sub-dimensions of vigor, devotion, and involvement (Table 3). Indeed, those who participate most frequently in MT activities experience higher levels of vigor, devotion, and emotional involvement.

Basically, from our analysis regarding the comparison within the MT Group, considering the frequency of participation in these activities, statistically significant differences were found among the healthcare workers. The results show that professionals who engage in MTAs more frequently (more than once a month) experience higher levels of resilience and engagement, and lower levels of psychological distress at work, specifically in the dimension of vigor, compared to those who participate in MT activities less frequently.

However, regarding objective c and as presented in Table 4, when analyzing psychological safety only within the MT Group, we find statistically significant differences both at a general level and in self-expression: during these activities, participants experience greater psychological safety overall and greater safety regarding self-expression compared to when they are engaged in other work activities. Also, when we compare job engagement within the MT Group, there are significant differences in all three sub-dimensions of the UWES scale. This means that participants who practice MT activities show higher levels of vigor, devotion, and emotional involvement during those activities when compared to other activities during their daily job in the healthcare department they belong to.

## 4. Discussion and Conclusions

Considering the data collected, the results indicate that the group of participants engaging in MTAs have higher levels of well-being, especially during these activities. Those who participate in MTAs have greater psychological well-being and less general dysphoria compared to those who do not participate in such activities, and there is a tendency among these operators to have better social functioning, greater job satisfaction, and lower levels of burnout. Our results are in line with a systematic review published by Djernis and colleagues (2019) [60] showing that nature-based mindfulness has a positive effect on psychological, physical, and social conditions. In addition, findings from meta-analyses suggest that exposure to natural settings improves attention and supports mental disengagement from daily pressures. Our results are consistent even with three studies’ results from Mayer and colleagues’ research (2009) [61]: the participants exposed to nature increased connectedness to nature, attentional capacity, positive emotions, and the ability to think critically about a life issue. Moreover, several experiences in nature such as photography, walking, or mindful engagement can strengthen feelings of connection to nature and, in some instances, with other people, thereby contributing to emotional well-being and prosocial behavior. Furthermore, exposure to green exercise (physical activity conducted in natural environments) improves self-esteem and mood, which are key indicators of mental health. Individuals with mental health challenges experienced some of the greatest gains in self-esteem [62].

This result confirms that good practices and positive organizational coexistence can contribute to improving overall quality of life, particularly in terms of psychophysical well-being [26]. Those who do not participate in MountainTherapy represents most individuals who fall into the intermediate level on the psychological well-being scale [50] (yellow), indicating a “warning sign”, which suggests a greater risk of developing issues related to psychological discomfort. This result aligns with recent studies [27,31,58,59,62] in the literature that explored the positive relationship between being in contact with nature and reduced cortisol levels, the stress hormone.

The differences that emerged between the two groups may also be related to other mechanisms that work alongside the role of the natural environment. Indeed, MountainTherapy is an activity that has other salient characteristics that distinguish it [63], such as the restructuring of the relationship between health professionals and patients, in a sense of equality and sharing, facilitated by the context in which everyone must face the difficulties and challenges typical of the mountainous environment; the reconfiguration of dynamics within the team, with a prevalence of cooperative and non-hierarchical relationships; and teamwork with individuals and entities not belonging to the healthcare sector (e.g., the Italian Alpine Club, mountain guides, etc.) that are involved due to their expertise and passion for the mountains. These elements indicate that there is greater autonomy for all professionals who carry out MTAs, resulting in their greater satisfaction, which in turn has a positive impact on their involvement in work and the organization, as job satisfaction, when considered globally, constitutes a protective factor against anxiety and depression, as well as against emotional burnout, which characterizes burnout [64]. As previously mentioned, burnout is located at the opposite end of the continuum associated with the concept of work engagement [65]. Therefore, a worker operating in conditions of well-being, who sees their professionalism valued and has the opportunity to participate in work processes, will be more strongly motivated and oriented towards greater commitment to their work and the company compared to an operator experiencing high levels of stress and burnout.

Considering now only the professionals who participate in MTAs, an analysis and comparison of the experiences during these activities versus other work activities reveal that they perceive higher levels of vigor, devotion, emotional involvement, and overall psychological safety and self-expression when engaged in MountainTherapy compared to when they are in the context of other work activities. This finding aligns with reports from other studies in the literature [66], which confirm the positive effects of nature-contact activities in enhancing vitality, energy, and quality of life for individuals. Furthermore, the beneficial effects we mentioned before are also linked to the frequency with which these activities are undertaken; indeed, participating in MountainTherapy more than once a month leads to greater engagement and personal well-being in the workplace regarding the dimension of vigor compared to those who engage in such activities only once a month.

In general, the data emerging from this multicentric research indicate that MountainTherapy could be a tool for promoting personal involvement and commitment in the organization and development of interpersonal relationships, playing a significant role in preventing burnout as well. In the end, our research can confirm the evidence present in the literature from studies that have analyzed psychological well-being and contact with nature [22,62,65].

In light of these results, therefore, we can consider it important to try to spread the practice of MountainTherapy, involving as many healthcare operators as possible. In this way, it would be possible to promote not only physical health, resulting from the physical activity enacted during MTAs, but also mental and psychological health and well-being, which is also evident from the various studies analyzing the impact of contact with nature. All this, with a view to the promotion of well-being and health on which Health Psychology is based, we refer to in our research practice. In conclusion, then, these results can be a good starting point for increasing the involvement of healthcare operators, in order to motivate them more to take part in MountainTherapy activities.

### Limitations and Future Projections

This study’s results provide a solid foundation for further investigations that go beyond the limitations of the current research, such as the small sample size and the selectivity and not random assignment between groups with which participants were recruited with possible self-selection bias. For this reason, these results cannot be generalized and are specific to the sample of this study. Furthermore, it is crucial to delve deeper into the psychological dimensions of interest, with a broader perspective that includes other professions, both healthcare-related and non-healthcare-related. In addition, an interesting aspect to consider concerns the possibility of conducting a study involving patients practicing MTAs, to comprehend what are the beneficial effects of these activities on those with mental distress as well.

Moreover, it seems important to investigate the difference within the MT Group to determine if the frequency of participation in MTAs could affect the psychological well-being outcomes of participants.

## Figures and Tables

**Figure 1 ijerph-22-01181-f001:**
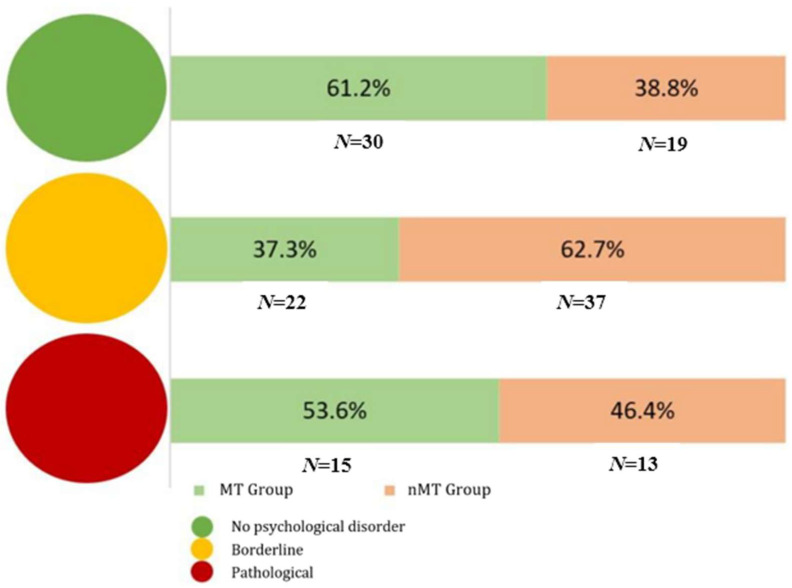
Distribution of MountainTherapy Participant Group (in green) and Non-MountainTherapy Participant Group (in orange) in the psychological distress traffic lights.

**Table 1 ijerph-22-01181-t001:** Sociodemographic characteristics of the MT Group and the nMT Group.

	MT Group (*n =* 83; 49.7%)		nMT Group (*n =* 84; 50.3%)		*χ*2
**Age** (years)					(*χ*2(39) = 39.88, *p* = 0.431)
mean (SD)	44.65 (11.32)		47.10 (9.86)		
	**Frequency (*n*)**	**Percentage (*%*)**	**Frequency (*n*)**	**Percentage (*%*)**	
**Gender**					(*χ2*(3) = 1.42, *p =* 0.701)
male	14	21.2	16	24.6	
female	49	74.2	46	70.8	
prefer not to answer	3	4.5	2	3.1	
other	/	/	1	1.5	
**Geographical location**					(*χ*2(3) = 1.798, *p* = 0.615)
north-east	42	50.6	40	47.6	
north-west	19	22.9	26	40.0	
center	14	16.9	10	11.9	
south	8	9.6	8	9.5	
islands	/	/	/	/	
**Referral Service**					
outpatient	20	32.3	28	45.9	
residential	18	29.0	13	21.3	
semi-residential	24	38.7	20	32.8	
**Years of employment**					(*χ*2(3) = 3.13, *p* = 0.373)
less than 5 years	23	34.8	15	23.1	
5 to 10 years	7	10.6	12	18.5	
10 to 20 years	14	21.2	16	24.6	
more than 20 years	22	33.3	22	33.8	
**Team role**					(*χ*2(6) = 1.87, *p* = 0.931)
social worker	2	3.0	1	1.5	
physician	3	4.5	2	3.1	
health and social worker (OSS)	4	6.1	5	7.7	
psychiatric rehabilitation	7	10.6	5	7.7	
therapist					
psychologist	11	16.7	8	12.3	
nurse	18	27.3	22	33.8	
professional educator	21	31.8	22	33.8	
**MT practice length**			/	/	
less than 1 year	8	12.1			
1 to 3 years	22	33.3			
3 to 5 years	14	21.2			
5 to 10 years	8	12.1			
more than 10 years	14	21.2			
**MT practice attendance**			/	/	
1 time per week	5	7.7			
1 time every 2 weeks	12	18.5			
1 time per month	47	72.3			
participation in visits	1	1.5			

**Table 2 ijerph-22-01181-t002:** Comparison between the MT Group and the nMT Group.

	M (SD)		t	*df*	*p*	*Cohen’s d*
	MT Group	nMT Group				
**GHQ-12 (Psychological distress)**						
total score	15.06 (6.31)	16.67 (4.38)	−1.721	117.282	0.044	−0.297
general dysphoria	9.04 (4.50)	10.20 (3.11)	−1.741	116.955	0.042	−0.300
social functioning	6.01 (2.43)	6.46 (1.92)	−1.195	134	0.117	−0.205
**MBI (Burnout)**						
emotional exhaustion	2.62 (1.12)	2.63 (0.86)	−0.117	127.295	0.454	−0.020
depersonalization	1.66 (0.74)	1.73 (0.72)	−0.578	137	0.282	−0.098
personal accomplishment	4.91 (0.67)	4.74 (0.79)	1.374	137	0.086	0.233
**UWES-9 (Job engagement)**						
vigor	3.51 (0.66)	3.36 (0.61)	1.343	133	0.091	0.231
devotion	4.06 (0.77)	3.98 (0.76)	0.633	133	0.264	0.109
emotional involvement	3.23 (0.60)	3.15 (0.64)	0.785	133	0.217	0.135
**RS-14 (Resilience)**	76.72 (13.86)	77.12 (13.30)	−0.172	133	0.432	
**Edmondson’s Scale** **(Psychological Safety)**						
general	5.00 (1.19)	5.01 (1.06)	−0.044	132	0.483	−0.007
self-expression	5.35 (1.09)	5.20 (1.00)	0.800	132	0.212	0.134

**Table 3 ijerph-22-01181-t003:** Comparison within the MT Group, with reference to the monthly frequency of MTAs.

	M (SD)		t	*df*	*p*	*Cohen’s d*
	MT Group More than Once a Month	MT Group Less than Once a Month				
**GHQ-12 (Psychological distress)**						
total score	12.71 (6.56)	15.96 (5.80)	1.921	63	0.030	6.00
general dysphoria	7.18 (4.84)	9.77 (4.10)	2.139	63	0.272	4.30
social functioning	5.53 (2.35)	6.19 (2.38)	0.984	63	0.164	2.37
**MBI (Burnout)**						
emotional exhaustion	2.39 (1.23)	2.67 (1.09)	0.886	63	0.190	1.12
depersonalization	1.31 (0.40)	1.73 (0.77)	2.857	54.107	0.003	0.70
personal accomplishment	5.00 (0.63)	4.89 (0.67)	−0.610	63	0.267	0.66
**UWES-9 (Job engagement)**						
vigor	3.80 (0.74)	3.39 (0.61)	−2.276	63	0.013	0.65
devotion	4.20 (0.83)	3.99 (0.74)	−0.954	63	0.175	0.77
emotional involvement	3.22 (0.78)	3.24 (0.55)	0.137	63	0.446	0.62
**RS-14 (Resilience)**	82.29 (11.735)	75.83 (12.309)	−1.882	63	0.032	12.17
**Edmondson’s Scale** **(Psychological Safety)**						
general	5.18 (1.04)	4.93 (1.25)	−0.714	63	0.239	1.20
self-expression	5.18 (1.26)	5.38 (1.03)	0.634	63	0.264	1.10

**Table 4 ijerph-22-01181-t004:** Comparison of variables within the MT Group, specifically focusing on the differences between MTA activities and other work activities.

	M (SD)		t	*df*	*p*	*Cohen’s d*
	MT Group During MT Activities	MT Group During Other Work Activities				
**UWES-9 (Job engagement)**						
vigor	5.88 (0.93)	3.50 (0.66)	−8.322	66	<0.001	0.86
devotion	6.26 (0.90)	4.06 (0.77)	−4.500	66	<0.001	0.88
emotional involvement	5.62 (1.06)	3.23 (0.60)	−8.322	66	0.002	1.01
**Edmondson’s Scale** **(Psychological Safety)**						
general	4.20 (0.61)	5.01 (1.19)	−5.819	66	<0.001	1.16
self-expression	4.35 (0.58)	5.35 (1.09)	−5.609	66	<0.001	1.03

## Data Availability

Data availability statements are available by request to authors.

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
