# Peer review of "The Relationship Between Well-Being and MountainTherapy in Practitioners of Mental Health Departments"

_ijerph, 2025, doi:10.3390/ijerph22081181_

Round 1

Reviewer 1 Report

Comments and Suggestions for Authors

Dear Authors and Editor, first of all, I would like to thank you for the opportunity to review your article.
First, I would like to say that this is an interesting study with many implications. However, I see some potential improvements.

In the introduction:
- One of the important weaknesses is the limited definition of psychological well-being; I think this is something that needs to be addressed.
- The gap in the literature regarding what this research seeks to address should be made much more explicit.
- I believe the article would be much more powerful if the research objectives and hypotheses were stated at the end of the introduction.

In materials and methods:
- I think the procedure is very well explained; furthermore, all the scales used are validated for the Italian context.
- One of the important aspects that needs to be resolved is how they grouped people in the MTA group, given that a single visit was enough to be considered in the MT group, as well as those who attended regularly. This is absolutely problematic, since grouping everyone in the same experimental group could underestimate the true effect. This can lead to an increase in internal variance.
To address this, perhaps they should create subcategories with high, medium, and low MT or control variability with a subgroup analysis based on frequency of participation or through an ANCOVA.
- Invariance should be calculated given that modified tests are required.
- Please state the overall internal consistency of all scales in addition to that of the factors.
- Please explicitly state the ethics committee that approved this research.
- Please state the sample size calculation explicitly.

Regarding the results:
- I think the results are presented in an orderly manner.
- Please present the effect sizes.
- Figure 1 raises many questions for me given how this "categorization" was done? The description of the instruments does not establish cutoff points either. How then is this categorization done?
- The results of the multigroup factorial invariance tests should be presented.

Regarding the discussion:
- It is the lowest part of the article.
- The results of this research need to be further elaborated by comparing them with the available scientific evidence.
- The limitations and projections of this research should be separately reported.
- Given the research design and presentation of results, it is impossible to attribute causality of TM to the psychological variables measured.
- These results cannot be generalized to other samples; therefore, they should be limited to the present sample.
- I do not observe an in-depth analysis of sociodemographic variables.

References
Review the reference format.
References should be reviewed, as several do not have a DOI.

Author Response

Dear Reviewers and Editor,

We are very grateful to you for reviewing the manuscript and providing valuable suggestions for improving our article. We have tried to consider all the feedback, addressed each of the questions/concerns that were raised (see below), and made changes to the manuscript following the recommendations where possible.     

Dear Authors and Editor, first of all, I would like to thank you for the opportunity to review your article.

First, I would like to say that this is an interesting study with many implications. However, I see some potential improvements.

Thank you very much for the interest shown in our research. We appreciate that our study is interesting to you.

In the introduction:

- One of the important weaknesses is the limited definition of psychological well-being; I think this is something that needs to be addressed.

Thank you for your comment. We added the definition of Ryff and Keyes (1995) from line 102 to 105: “Ryff and Keyes (1995) described psychological well-being as a multidimensional concept that includes several key elements like self-acceptance, positive relationships, environmental control, autonomy or purpose in life. These factors are crucial for maintaining a healthy psychological state”.

- The gap in the literature regarding what this research seeks to address should be made much more explicit.

Thank you for your suggestion. We decided to add a specific sentence to make it more clear in lines 198 - 199: “Based on that, it is evident that the literature reveals a gap in exploring the influence of nature on the well-being of health professionals”.

- I believe the article would be much more powerful if the research objectives and hypotheses were stated at the end of the introduction.

Thank you for your comment. Also considering the suggestion of other reviewers, we decided to add a new paragraph at the end of the Introduction section 1.1 Aims of the study. We modified this paragraph adding the aim of the study at the beginning from line 201 to 212: “The study seeks to investigate well-being levels of health professionals employed in Italian Departments of Mental Health who regularly engage in MTA and those who do not. 

Furthermore, it hypothesizes three different objectives: (objective a) professionals involved in Mountain Therapy activities exhibit higher psychological well-being, resilience, psychological safety, engagement, and workplace well-being and lower levels of burnout risk compared to those who do not practice these activities; (objective b) the frequency of these activities can positively influence the psychological well-being levels of participants; and (objective c) MT group will achieve higher well-being scores when MT activities are compared to other workplace activities (e.g., doing therapy and visits patients in the facility).

In materials and methods:

- I think the procedure is very well explained; furthermore, all the scales used are validated for the Italian context.

Thank you for your comment. We appreciate it. 

- One of the important aspects that needs to be resolved is how they grouped people in the MTA group, given that a single visit was enough to be considered in the MT group, as well as those who attended regularly. This is absolutely problematic, since grouping everyone in the same experimental group could underestimate the true effect. This can lead to an increase in internal variance.

To address this, perhaps they should create subcategories with high, medium, and low MT or control variability with a subgroup analysis based on frequency of participation or through an ANCOVA.

Thank you so much for your suggestion. According to your other successive comment, we decided to run the ANCOVA to compare the three groups (who practice MT activities less than once a month, more than once a month and those who don’t practice these activities). We controlled the sociodemographic variables, as you suggested in your following comment, including: age, gender, years of service, and MT frequencies (who practice MT activities less than once a month, more than once a month and those who don’t practice these activities).

Specifically, we found that our results are also confirmed in this case, so doing MT activities and doing them regularly help in improving the psychological well-being of the participants of our research (as shown in the table below). The significance, therefore, remains for the two dimensions of the GHQ-12 (total score and general dysphoria).

Comparison between MT Group, which practices MTA less than once a month, MT Group, which practices MTA more than once a month and nMT Group 

M (SD)

F

df

p

MT Group < once a month 

MT Group > once a month 

nMT Group

GHQ-12 (Psychological distress) 

 total score

 general dysphoria

social functioning

15.75 (5.24)

9.68 (3.80)

6.07 (2.19)

12.14 (6.57)

6.86 (5.01)

5.29 (2.27)

16.69 (4.59)

10.27 (3.27)

6.42 (1.95)

2.765

3.119

.985

11, 105

11, 105

11, 105

.022

.011

.430

MBI (Burnout)

emotional exhaustion

depersonalization

personal accomplishment

2.67 (1.05)

1.71 (0.72)

4.87 (0.69)

2.63 (1.33)

1.24 (0.31)

5.08 (0.64)

2.62 (0.88)

1.71 (0.74)

4.79 (0.80)

.202

1.308

1.245

11, 105

11, 105

11, 105

.961

.266

.293

UWES-9 (Job engagement)

 vigor

 devotion

 emotional involvement

3.38 (0.59)

4.05 (0.70)

3.21 (0.56)

3.80 (0.75)

4.39 (0.80)

3.14 (0.82)

3.39 (0.63)

4.01 (0.74)

3.18 (0.64)

1.347

1.039

.338

11, 105

11, 105

11, 105

.250

.398

.889

RS-14 (Resilience)

75.73 (12.00)

83.29 (9.94)

77.41 (13.65)

.826

11, 105

.534

Edmondson’s Scale (Psychological Safety)

general

 self-expression

4.97 (1.21)

5.43 (1.04)

5.09 (1.02)

5.20 (1.21)

5.03 (1.06)

5.24 (.98)

.562

1.162

11, 105

11, 105

.729

.332

As you can see in the table below, the sociodemographic characteristics we controlled for don’t have any effect on the dependent variables.

Anyway we are conscious about your suggestion and we really appreciate it, but at this moment we have decided to not include these results in our article because the study design at the beginning was different (comparing MT group and nMT group) and because to proceed with the comparison also within the MT group we should aim to have at least 30 people in each group (in the present study we have 69 who don’t practice MTA, 48 who practice lees that once a month and 17 who practice more than once a month). If you consider that it is better to insert these results (e.g., as a note in the article), we will do it. Of course, we decided to take into account this valuable suggestion for future research on how the frequency can affect the outcomes of our participants. 

- Invariance should be calculated given that modified tests are required.

Thank you very much for your thoughtful consideration. We would like to kindly clarify that utilizing the questionnaires outside of their original context was not part of the primary aims of our study. However, we believe that the existing theoretical framework provides a solid foundation for this approach. We have included references to the use of the questionnaires in different settings than initially foreseen. For example, we refer to the insightful study by Vecina et al. (2012) (Vecina, Chacón, Sueiro, & Barrón, (2012). Volunteer engagement: Does engagement predict the degree of satisfaction among new volunteers and the commitment of those who have been active longer?  Applied psychology, 61(1), 130-148).

In our future research, we would be delighted to explore this perspective further. Additionally, we would like to underline that we are the same authors currently engaged in validating the Psychological Safety scale in Italian and can confidently confirm the internal validity of the questionnaire through Confirmatory Factor Analysis (CFA).

- Please state the overall internal consistency of all scales in addition to that of the factors.

Thank you for your comment. As discussed by Nunally & Bernstein (1994) and Cohen (2010) when a test includes multiple factors, calculating alpha for the overall test may not be appropriate. Therefore, it is more meaningful to compute alpha for each construct rather than for the entire test or scale You can find more information in the following articles:

  • Nunnally J, Bernstein L. Psychometric theory. New York: McGraw-Hill Higher, INC; 1994. 
  • Cohen R, Swerdlik M. Psychological testing and assessment. Boston: McGraw-Hill Higher Education; 2010.

Additionally, for a more comprehensive understanding, we have included all the Cronbach’s alphas for the tests we used in our study. 

To answer to your question, we report here the overall Alphas’ values for the different scales used: 

  • GHQ= .878
  • MBI= .720
  • UWES= .898
  • UWES MT GROUP= .961
  • RS= .936
  • PSYCHOLOGICAL SAFETY= .819
  • PSYCHOLOGICAL SAFETY MT GROUP= .740

- Please explicitly state the ethics committee that approved this research.

Thank you for the suggestion. We added the approval from the Committee for Research Integrity and Ethics of the University of Bergamo at line 226 and from line 570 to line 574.  

- Please state the sample size calculation explicitly.

Thank you for your comment. We have included the sample sizes for both groups in our study, from line 233 to 238. However, we would like to emphasize that it was quite challenging to gather a substantial number of participants for each group (those who practice MT activities and those who do not) so we decided to define our effect size as 0.5. We also added this point from line 232 to 233 in our article. 

Regarding the results:

- I think the results are presented in an orderly manner.

Thank you for your comment

- Please present the effect sizes.

Thank you for your comment. We added the Cohen d in the Table 2 starting from line 418. 

- Figure 1 raises many questions for me given how this "categorization" was done? The description of the instruments does not establish cutoff points either. How then is this categorization done?

Thank you for your valuable feedback. We appreciate your insights. In the initial version of our manuscript, we provided a detailed explanation of the categorization in section 2.3.1, which pertains to the description of the General Health Questionnaire [GHQ-12], Italian version from line 253 to 275. We also included the relevant reference that informed our analyses (Giorgi G, Perez JML, D’Antonio AC, et al. The general health questionnaire (GHQ-12) in a sample of Italian workers: mental health at individual and organizational level. World J Med Sci. 2014;11(1):47–56.)

- The results of the multigroup factorial invariance tests should be presented.

Thank you for your comment. We appreciate your observation regarding the potential impact of participation frequency in MT activities. But as we said before, this aspect wasn’t part of the primary aims of our study; we acknowledge this important aspect and plan to consider it in future research. We have also included this point in the limitations and future directions section of the article to highlight the importance of exploring this topic further.

Regarding the discussion:

- It is the lowest part of the article.

Thank you for this consideration. We have incremented the information in the discussion section, to make it more clear, incisive, and detailed.

- The results of this research need to be further elaborated by comparing them with the available scientific evidence.

Thank you for your comment. Following your suggestion and other reviewer’s suggestions, we decided to add in the discussion section from line 479 to 488 this paragraph considering different scientific evidences related to our results: “Our results are consistent even with three studies’s results by Mayer and colleagues research (2009) [61]: the participants exposed to nature increased connectedness to nature, attentional capacity, positive emotions, and the ability to think critically about a life issue. Moreover, several experiences in nature such as photography, walking, or mindful engagement can strengthen feelings of connection to nature and, in some instances, with other people, thereby contributing to emotional well-being and prosocial behavior. Furthermore, exposure to green exercise (physical activity conducted in natural environments) improves self-esteem and mood which are key indicators of mental health. Individuals with mental health challenges experienced some of the greatest gains in self-esteem [62]”.

- The limitations and projections of this research should be separately reported.

Thank you for your consideration. We added the 4.1 paragraph Limitations and future projections from line 547 to 560: “This study results provide a solid foundation for further investigations that go beyond the limitations of the current research, such as the small sample size and the selectivity and not random assignment between groups with which participants were recruited with possible self-selection bias. For this reason these results cannot be generalized and are specific to the sample of this study. Furthermore, it is crucial to delve deeper into the psychological dimensions of interest, with a broader perspective that includes other professions, both healthcare-related and non-healthcare-related. In addition, an interesting aspect to consider concerns the possibility of conducting a study involving patients practicing MountainTherapy activities, to comprehend what are the beneficial effects of these activities on those with mental distress as well.

Moreover, it seems important to investigate the difference within the MT group to determine if the frequency of participation in MT activities could affect the psychological well-being outcomes of participants.”

- Given the research design and presentation of results, it is impossible to attribute causality of TM to the psychological variables measured.

Thank you for this point. We have changed in the manuscript all the possible terms related to a causality relationship. Also, from line 551 to line 552 we added “results cannot be generalized and are specific to the sample of this study.”.

- These results cannot be generalized to other samples; therefore, they should be limited to the present sample.

Thank you for your comment. We decided to add the sentence related to the impossibility to generalize the results in the Limitations and future projections paragraph, in particular in the lines 551 and 552 “For this reason these results cannot be generalized and are related only to the present study sample”. 

- I do not observe an in-depth analysis of sociodemographic variables.

Thank you for your comment. We have done this analysis and commented on the results in your previous suggestion.

References
Review the reference format.

Thank you for the comment. We changed the reference format, using the correct one.

References should be reviewed, as several do not have a DOI.

We added all the DOIs, where possible.

Reviewer 2 Report

Comments and Suggestions for Authors

The title and abstract are adequate; however, the study design should be incorporated into the abstract. Authors concluded that MTA seem to be an effective intervention for promoting psychological health. Instead, suggest using “MTA can be considered a valid intervention for the promotion of mental health”. The phrase 'seem to be' is too tentative for a study conclusion and may weaken the perceived strength of the findings.

The introduction is well-written and supported by an adequate number of relevant references. The background effectively establishes the rationale for the study, making its conduct well-justified.

The authors should specify the study design in the methodology section. The research instruments are clearly described and appropriately referenced. However, ethical considerations should also be addressed in the manuscript.

The tables and figures presented in the results section are clear and easy to interpret. While the Chi-square test can indicate an association between two variables, it does not assess differences between them. Therefore, in lines 403 and 414 of the results section, the term 'significant difference' should be replaced with 'significant association.' Additionally, the distribution pattern of the data has not been described, and the authors should provide justification for the use of the t-test in Tables 2 and 3.

Both the discussion and conclusion are appropriate and align well with the results presented. The authors are advised to replace the phrase 'seem to be' with a more assertive term in the conclusion.

Comments on the Quality of English Language

The overall language quality is adequate, though a light proofreading could further improve clarity.

Author Response

Dear Reviewers and Editor,

We are very grateful to you for reviewing the manuscript and providing valuable suggestions for improving our article. We have tried to consider all the feedback, addressed each of the questions/concerns that were raised (see below), and made changes to the manuscript following the recommendations where possible.     

Dear reviewer, we thank you for your interest in our research and also for your advice on improving the structure of the article. We reproduce below your comments with our responses and additions made following your advice.

The title and abstract are adequate; however, the study design should be incorporated into the abstract. Authors concluded that MTA seem to be an effective intervention for promoting psychological health. Instead, suggest using “MTA can be considered a valid intervention for the promotion of mental health”. The phrase 'seem to be' is too tentative for a study conclusion and may weaken the perceived strength of the findings.

Thank you for your consideration. We added the study design in the abstract in lines 16 - 19: “This cross-sectional study aims to explore the levels of well-being among professionals working in Italian Departments of Mental Health who regularly engage in MTA and those who do not”. 

In addition, following your suggestion, we rephrased the sentence in lines 35- 36 “MTA can be considered a valid intervention for the promotion of mental health of healthcare workers”.

The introduction is well-written and supported by an adequate number of relevant references.

Thank you for your comment. 

The background effectively establishes the rationale for the study, making its conduct well-justified.

Thank you for your comment. 

The authors should specify the study design in the methodology section. 

Thank you for comment. We added the specific study design in the Design and Sampling paragraph in the line 216 and 217: “The present study has been designed as cross-sectional, allowing the investigation in a specific moment in time.”

The research instruments are clearly described and appropriately referenced. 

Thank you for your comment. 

However, ethical considerations should also be addressed in the manuscript.

Thank you for the suggestion. We added the approval from the Committee for Research Integrity and Ethics of the University of Bergamo from line 570 to line 574 of our paper. 

The tables and figures presented in the results section are clear and easy to interpret. 

Thank you for your comment. 

While the Chi-square test can indicate an association between two variables, it does not assess differences between them. Therefore, in lines 403 and 414 of the results section, the term 'significant difference' should be replaced with 'significant association.' 

Thank you for your comment. We modified the term difference with association in the paragraph from line 396 to 404: “The bivariate analyses showed that the MT Group and the nMT Group did not exhibit statistically significant association for any of the following variables: gender (χ2(3, N=131)=1.42, p=.701), age (χ2(39, N=123)=39.88, p=.431), affiliation service (χ2(2, N=123)=2.50, p=.287), geographical origin (χ2(3, N=167)=1.798, p=.615), length of service (χ2(3, N=131)=3.13, p=.373) and role (χ2(6, N=131)=1.87, p=.931).

Concerning the variables of interest, we observed several statistically significant associations and noteworthy trends toward significance, both in the comparison between the two groups and within the MT Group itself. Comparing the two groups, it was found that in psychological distress - measured using the GHQ-12 scale - those who do not participate in MountainTherapy activities have, on average, higher scores both at the total level and in the sub-dimension of general dysphoria”

Additionally, the distribution pattern of the data has not been described, and the authors should provide justification for the use of the t-test in Tables 2 and 3.

Thank you for your comment. We added in the Statistical Data Analysis paragraph what we’ve run to define the normal distribution of the scale we used from line 373 to 376: “The dimensions of the scale used for the study were preliminarily submitted to analyses to check the normal distribution by calculating mean, standard deviation, and indices of skewness and kurtosis; West et al. (1995) [62] recommend concern if skewness > |2| and kurtosis > | 7|.”

We also added in the Result section we added the information about the distribution of our dimensions from line 393 to 398 “First of all, considering the West et al. (1995) [62] recommendations, all the dimensions showed an acceptable distribution (SkewnessMIN = –0.82 - SkewnessMAX = 1.22; KurtosisMIN = –0.96-KurtosisMAX = 2.56)”.

Both the discussion and conclusion are appropriate and align well with the results presented. 

Thank you for your comment. 

The authors are advised to replace the phrase 'seem to be' with a more assertive term in the conclusion.

Thank you for your comment. We replaced the “seem to” with “indicate” in the discussion and conclusion section. 

Reviewer 3 Report

Comments and Suggestions for Authors

The topic presented in the manuscript is quite interesting and relevant. Especially when it comes to mental health professionals who face high stress and the possibility of burnout in their work. Therefore, it is important to research and understand methods or therapies that can reduce the occurrence of these difficulties. However, the manuscript has several errors that would be important to correct.

Comments 1. The summary should present the results of the calculations, not just describe them in words.

Comments 2. The introduction is overly lengthy. It may be beneficial to think about ways to shorten it while still communicating the significance and relevance of the subject. I believe it’s important to emphasize the following structure: First, outline the physical and mental challenges that healthcare professionals encounter (such as overwork, burnout, and excessive workloads). Second, it addresses the need for support for these individuals. Third, explain what Mountain Therapy is, identify the practitioners involved, and describe how the therapy is performed. Fourth, highlight the benefits and positive traits of Mountain Therapy.

Comments 3. The paragraph from line 99 to line 111 could be used in the Discussion.

Comments 4. Paragraphs from lines 112 to 130 are repetitive in content. It may be beneficial to reconsider how this information can be used, such as in a discussion.

Comments 5. The research objectives and hypotheses should be placed in a separate paragraph titled "Aim of the study."

Comments 6. The two aims are quite similar, so a more detailed explanation is needed for readers to understand their differences clearly.

Comments 7. There is a lack of consistency and clarity in how respondents were recruited. Did the research team approach hospitals or other healthcare facilities to find respondents? Were respondents recruited through Mountain Therapy projects? How were professionals who are not involved in MT recruited?

Comments 8. In line 277, the paragraph title should be italic. There should be no citation here. References [50] and [51] should be included in the text. Generally, I suggest renaming section 2.2.1 to 2.3 Questionnaire.

Comments 9. Same notes 2.2.2, 2.2.3, 2.2.4, 2.2.5.

Comments 10. I think the information from lines 402-406 should be included in table 1.

Comments 11. Figure No. 1 should present not only percentages, but also individuals (n).

Comments 12. In my assessment, the authors employed a substantial number of questionnaires, thus obtaining significant data that clarifies the advantages of MT. I recommend the performance of logistic regression analysis to further examine and interpret the collected data.

Comments 13. The discussion section is currently incomplete. In this context, it is imperative to justify and analyze the results, providing thorough explanations and drawing comparisons with the works of other researchers in the field.

Comments 14. The study does not provide a clear conclusion, which should be emphasized in Paragraph 5.

Comments 15. I recommend addressing the limitations associated with this study. It is essential to highlight that the study employs a cross-sectional design, which typically does not allow for the establishment of causal relationships.

I hope my insights will help you!

Author Response

Dear Reviewers and Editor,

We are very grateful to you for reviewing the manuscript and providing valuable suggestions for improving our article. We have tried to consider all the feedback, addressed each of the questions/concerns that were raised (see below), and made changes to the manuscript following the recommendations where possible.     

The topic presented in the manuscript is quite interesting and relevant. Especially when it comes to mental health professionals who face high stress and the possibility of burnout in their work. Therefore, it is important to research and understand methods or therapies that can reduce the occurrence of these difficulties. However, the manuscript has several errors that would be important to correct.

Dear reviewer, we thank you for your interest in our research and also for your advice on improving the structure of the article. We reproduce below your comments with our responses and additions made following your advice. 

Comments 1. The summary should present the results of the calculations, not just describe them in words.

Thank you for your comment. We added in the Abstract some results related to the t-test from line 30 to 35. 

Comments 2. The introduction is overly lengthy. It may be beneficial to think about ways to shorten it while still communicating the significance and relevance of the subject. I believe it’s important to emphasize the following structure: First, outline the physical and mental challenges that healthcare professionals encounter (such as overwork, burnout, and excessive workloads). Second, it addresses the need for support for these individuals. Third, explain what Mountain Therapy is, identify the practitioners involved, and describe how the therapy is performed. Fourth, highlight the benefits and positive traits of Mountain Therapy.

Thank you for this suggestion. We have shortened the introduction section by following this pattern of topic presentation.

Comments 3. The paragraph from line 99 to line 111 could be used in the Discussion.

Thank you for your suggestion. We decided to move this paragraph to the discussion section from line 474 also considering other additional information from other reviewers. 

Comments 4. Paragraphs from lines 112 to 130 are repetitive in content. It may be beneficial to reconsider how this information can be used, such as in a discussion.

Thank you for this suggestion. We have reformulated the introduction section to make it more clear and without repetitive parts.

Comments 5. The research objectives and hypotheses should be placed in a separate paragraph titled "Aim of the study."

Thank you for your comment. We decided to add a new paragraph at the end of Introduction section 1.1 Aims of the study. We modified this paragraph adding the aim of the study and the beginning of the paragraph from line 201 to 204: “The study seeks to investigate well-being levels of health professionals employed in Italian Departments of Mental Health who regularly engage in MTA and those who do not”.

Comments 6. The two aims are quite similar, so a more detailed explanation is needed for readers to understand their differences clearly.

Thank you for your consideration. We rephrased the paragraph related to the aims and objectives of the study from line 201 to 212:  “The study seeks to investigate well-being levels of health professionals employed in Italian Departments of Mental Health who regularly engage in MTA and those who do not. 

Furthermore, it hypothesizes three different objectives: (objective a) professionals involved in Mountain Therapy activities exhibit higher psychological well-being, resilience, psychological safety, engagement, and workplace well-being and lower levels of burnout risk compared to those who do not practice these activities; (objective b) the frequency of these activities can positively influence the psychological well-being levels of participants; and (objective c) MT group will achieve higher well-being scores when MT activities are compared to other workplace activities (e.g., doing therapy and visits patients in the facility).”

Comments 7. There is a lack of consistency and clarity in how respondents were recruited. Did the research team approach hospitals or other healthcare facilities to find respondents? Were respondents recruited through Mountain Therapy projects? How were professionals who are not involved in MT recruited?

Thank you for your valuable feedback. We provided details about our participants on lines 221 to 223, where we mentioned that: “Participants are all those who join the MTA programs offered by their department, with varying frequency: once a week, once every two weeks, once a month, or participation in visits" Additionally, from lines 227 to 231, we explained our recruitment process, stating that "Participants were recruited through convenience sampling. Participating departments were requested to identify all healthcare professionals who engaged in MountainTherapy activities (MT Group) during 2023, as well as another group of professionals who did not perform MountainTherapy but belonged to the same services and shared similar characteristics as the participants (nMT Group)”.

Comments 8. In line 277, the paragraph title should be italic. There should be no citation here. References [50] and [51] should be included in the text. Generally, I suggest renaming section 2.2.1 to 2.3 Questionnaire.

Thank you for your suggestion. We modified in italics the title and also the section numbers. We also included the references in the text in the lines 254 - 255: “General Health Questionnaire [GHQ-12] [48], Italian version of Piccinelli e Politi [49] has been designed to identify potential warning signs regarding psychological distress”.

Comments 9. Same notes 2.2.2, 2.2.3, 2.2.4, 2.2.5.

Thank you for your comment. We appreciate your point of view so, as previously, we modified the title included the references in the text for all the paragraphs:

  • 2.2.2 is now 2.3.2 and we included the references in the lines 277 - 278 “The Maslach Burnout Inventory [MBI] [51], Italian version of di Sirigatti e Stefanile [52] is aimed at investigating burnout syndrome”.
  • 2.2.3 is now 2.3.3 and we included the references from line 300 “Utrecht Work Engagement Scale [UWES-9] [53], Italian version of Balducci et al. [54] consists of 17 items for which the participant must assess - on a 7-point Likert scale (from 0 = never to 6 = daily)......”. 
  • 2.2.4 is now 2.3.4 and we included the references from the lines 326 to 328 “Resilience Scale [RS-14] [55], Italian version of Callegari et al. [56] is designed to investigate individuals’ resilience and the resources they have available to effectively cope with stressful or traumatic events, face difficulties, and adapt to change”.
  • 2.2.5 is now 2.3.5 and we included the references from the line 343 to 346 “Edmondson’s team psychological safety scale [57], the Italian version will be published by Todaro et al. [58] is a scale that measures the psychological safety perceived by individuals within their work context - through responses to 7 items on a 7-point Likert scale (from 1 = strongly disagree to 7 = strongly agree).....”.

Comments 10. I think the information from lines 402-406 should be included in table 1.

Thank you for your comment. We added the information related to the chi-squadre in the Table 1. 

Comments 11. Figure No. 1 should present not only percentages, but also individuals (n).

Thank you for your comment. We added the N on Figure 1. 

Comments 12. In my assessment, the authors employed a substantial number of questionnaires, thus obtaining significant data that clarifies the advantages of MT. I recommend the performance of logistic regression analysis to further examine and interpret the collected data.

Thank you for your comment. We would like to clarify that the original design of our study was focused on comparing the Mountain Therapy (MT) group with a control group of individuals who do not engage in Mountain Therapy activities. This comparison also takes into account the frequency of participation in the MT activities for those who do practice them.

As such, we opted to utilize t-test analysis for our comparisons. In this framework, participation in Mountain Therapy Activities (MTA) doesn’t rely on the levels across the various tests proposed. We hope this explanation clarifies our methodological choices.

In addition and considering your comment and the suggestion of another reviewer, we decided to run an ANCOVA analysis controlling the effect of gender, age, MTA frequency and length of service. Specifically, we found that our results are confirmed, so doing MT activities and doing them regularly help in improving the psychological well-being of the participants of our research (as shown in the table below). The significance, therefore, remains for the two dimensions of the GHQ-12 (total score and general dysphoria).

Comparison between MT Group, which practices MTA less than once a month, MT Group, which practices MTA more than once a month and nMT Group 

M (SD)

F

df

p

MT Group < once a month 

MT Group > once a month 

nMT Group

GHQ-12 (Psychological distress) 

 total score

 general dysphoria

social functioning

15.75 (5.24)

9.68 (3.80)

6.07 (2.19)

12.14 (6.57)

6.86 (5.01)

5.29 (2.27)

16.69 (4.59)

10.27 (3.27)

6.42 (1.95)

2.765

3.119

.985

11, 105

11, 105

11, 105

.022

.011

.430

MBI (Burnout)

emotional exhaustion

depersonalization

personal accomplishment

2.67 (1.05)

1.71 (0.72)

4.87 (0.69)

2.63 (1.33)

1.24 (0.31)

5.08 (0.64)

2.62 (0.88)

1.71 (0.74)

4.79 (0.80)

.202

1.308

1.245

11, 105

11, 105

11, 105

.961

.266

.293

UWES-9 (Job engagement)

 vigor

 devotion

 emotional involvement

3.38 (0.59)

4.05 (0.70)

3.21 (0.56)

3.80 (0.75)

4.39 (0.80)

3.14 (0.82)

3.39 (0.63)

4.01 (0.74)

3.18 (0.64)

1.347

1.039

.338

11, 105

11, 105

11, 105

.250

.398

.889

RS-14 (Resilience)

75.73 (12.00)

83.29 (9.94)

77.41 (13.65)

.826

11, 105

.534

Edmondson’s Scale (Psychological Safety)

general

 self-expression

4.97 (1.21)

5.43 (1.04)

5.09 (1.02)

5.20 (1.21)

5.03 (1.06)

5.24 (.98)

.562

1.162

11, 105

11, 105

.729

.332

Anyway we are conscious about your suggestion and we really appreciate it, but at this moment we have decided to not include these results in our article because the study design at the beginning was different (comparing MT group and nMT group) and because to proceed with the comparison also within the MT group we should aim to have at least 30 people in each group (in the present study we have 69 who don’t practice MTA, 48 who practice lees that once a month and 17 who practice more than once a month). If you consider that it is better to insert these results (e.g., as a note in the article), we will do it.  Of course, we decided to take into account this valuable suggestion for future research.

Comments 13. The discussion section is currently incomplete. In this context, it is imperative to justify and analyze the results, providing thorough explanations and drawing comparisons with the works of other researchers in the field.

Thank you for your comment. We decided to depeen better the paragraph related to the discussion and conclusion. 

Comments 14. The study does not provide a clear conclusion, which should be emphasized in Paragraph 5.

Thank you for this comment. We have detailed the discussion, including a conclusion connected to future researchers and to the practical implications of the research.

Comments 15. I recommend addressing the limitations associated with this study. It is essential to highlight that the study employs a cross-sectional design, which typically does not allow for the establishment of causal relationships.

Thank you for your consideration. We decided to add the 4.1 paragraph related to the limitations and future projections from line 548 to 560: “This study results provide a solid foundation for further investigations that go beyond the limitations of the current research, such as the small sample size and the selectivity and not random assignment between groups with which participants were recruited with possible self-selection bias. For this reason these results cannot be generalized and are specific to the sample of this study. Furthermore, it is crucial to delve deeper into the psychological dimensions of interest, with a broader perspective that includes other professions, both healthcare-related and non-healthcare-related. In addition, an interesting aspect to consider concerns the possibility of conducting a study involving patients practicing MountainTherapy activities, to comprehend what are the beneficial effects of these activities on those with mental distress as well.

Moreover, it seems important to investigate the difference within the MT group to determine if the frequency of participation in MT activities could affect the psychological well-being outcomes of participants.” and we added the study design in the the Design and Sampling paragraph in the line 216 and 217 “The present study has been designed as cross-sectional, allowing the investigation in a specific moment in time”.

I hope my insights will help you!

Yes, thank you so much. Your insights helped us improve the quality of our article.

Round 2

Reviewer 1 Report

Comments and Suggestions for Authors

Dear authors,  

Thank you for the improvements made to the manuscript. From my point of view, it is now suitable for publication. Congratulations!

Author Response

Dear authors,  

Thank you for the improvements made to the manuscript. From my point of view, it is now suitable for publication. Congratulations!

Dear reviewer,

Thank you so much for your comment. We greatly appreciate that the manuscript is now improved, thanks mainly to the revisions received.

Reviewer 3 Report

Comments and Suggestions for Authors

Thank you for submitting the corrected version of your manuscript. I appreciate the effort you have put into addressing the previous comments, and I find your research both interesting and valuable.

However, I do have a few additional comments and suggestions that I believe could further improve the quality of the manuscript.

Comments 1. The abstract is too long; it could be shortened while preserving the main idea.

Comments 2. I suggest dividing the introduction into separate paragraphs and creating titles. e.g. 1.1..... 1.2........

Comments 3.  In my opinion, the 2.3.5 Edmondson Team Psychological Safety Scale [60], the Italian version of which will be published by Todaro et al. [61], should be simply the 2.3.5 Edmondson Team Psychological Safety Scale. And citations of the scales should be in the text.

Author Response

Thank you for submitting the corrected version of your manuscript. I appreciate the effort you have put into addressing the previous comments, and I find your research both interesting and valuable. However, I do have a few additional comments and suggestions that I believe could further improve the quality of the manuscript.

Thank you for appreciating the changes made based on the suggestions received. We are glad that the manuscript is now clearer and more interesting. Thank you also for the additional comments, which allow us to further refine the structure of the manuscript.

Comments 1. The abstract is too long; it could be shortened while preserving the main idea.

Response 1: Thank you for your suggestion. We’ve reduced the abstract with 196 words in total. The new version of the abstract is structured as below:

Background. Healthcare workers’ health can be influenced by physical, psychological, social, emotional and work-related stress. MountainTherapy Activities (MTA) are an integrated therapeutic approach that uses nature to enhance their well-being through group activities like hiking. This cross-sectional study examines well-being levels among Italian Departments of Mental Health workers who do or don't participate in MTA. It hypothesizes that MTA may reduce burnout, boost psychological resilience, and increase job satisfaction. Methods. The study involved 167 healthcare workers from 11 Italian Local Health Authorities, divided into MTA (who take part in MTA; N=83) and non-MTA (who have never participated in MTA; N=84) groups. They completed five validated questionnaires on psychological distress, burnout, resilience, job engagement, and psychological safety. Data was compared between groups, considering MTA frequency and well-being differences during MTAs versus workplace activities. Results. MTA participants scored higher in psychological well-being (t(117.282)=-1.721, p=.044) and general dysphoria (t(116.955)=-1.721, p=.042). Additionally, during MTA, they showed greater job engagement (vigor: t(66)=-8.322, p<.001; devotion: t(66)=-4.500, p<.001; emotional involvement: t(66)=-8.322, p=.002) and psychological safety (general: t(66)=-5.819, p<.001; self-expression: t(66)=-5.609, p<.001) compared to other activities. Conclusions. MTA can be considered a valid intervention for the promotion of the mental health of healthcare workers.

Comments 2. I suggest dividing the introduction into separate paragraphs and creating titles. e.g. 1.1..... 1.2........

Response 2: Thank you for the suggestion. We have divided the introduction into paragraphs: 1.1 Work-related stress and burnout in health care services; 1.2 Theoretical models and psychological well-being in the work context; 1.3 Environmental and MountainTherapy Activities; 1.4 Aims of the study.

Comments 3.  In my opinion, the 2.3.5 Edmondson Team Psychological Safety Scale [60], the Italian version of which will be published by Todaro et al. [61], should be simply the 2.3.5 Edmondson Team Psychological Safety Scale. And citations of the scales should be in the text.

Response 3: Thank you for the suggestion. We have made the requested changes.